# Improving the Generalization of Adversarial Training with Domain Adaptation

**Chuanbiao Song**
Department of Computer Science
Huazhong University of Science and Technology
Wuhan 430074, China
cbsong@hust.edu.cn

**Kun He**[*]
Department of Computer Science
Huazhong University of Science and Technology
Wuhan 430074, China
brooklet60@hust.edu.cn

**Liwei Wang**
Department of Machine Intelligence
Peking University
wanglw@pku.edu.cn

**John E. Hopcroft**
Department of Computer Science
Cornell University
Ithaca 14850, NY, USA
jeh@cs.cornell.edu

## Abstract

By injecting adversarial examples into training data, adversarial training is promising for improving the robustness of deep learning models. However, most existing adversarial training approaches are based on a specific type of adversarial attack. It may not provide sufficiently representative samples from the adversarial domain, leading to a weak generalization ability on adversarial examples from other attacks. Moreover, during the adversarial training, adversarial perturbations on inputs are usually crafted by fast single-step adversaries so as to scale to large datasets. This work is mainly focused on the adversarial training yet efficient FGSM adversary. In this scenario, it is difficult to train a model with great generalization due to the lack of representative adversarial samples, aka the samples are unable to accurately reflect the adversarial domain. To alleviate this problem, we propose a novel Adversarial Training with Domain Adaptation (ATDA) method. Our intuition is to regard the adversarial training on FGSM adversary as a domain adaption task with limited number of target domain samples. The main idea is to learn a representation that is semantically meaningful and domain invariant on the clean domain as well as the adversarial domain. Empirical evaluations on Fashion-MNIST, SVHN, CIFAR-10 and CIFAR-100 demonstrate that ATDA can greatly improve the generalization of adversarial training and the smoothness of the learned models, and outperforms state-of-the-art methods on standard benchmark datasets. To show the transfer ability of our method, we also extend ATDA to the adversarial training on iterative attacks such as PGD-Adversial Training (PAT) and the defense performance is improved considerably.

## 1 Introduction

Deep learning techniques have shown impressive performance on image classification and many other computer vision tasks. However, recent works have revealed that deep learning models are often vulnerable to adversarial examples (Szegedy et al., 2014; Goodfellow et al.; Papernot et al., 2016), which are maliciously designed to deceive the target model by generating carefully crafted adversarial perturbations on original clean inputs. Moreover, adversarial examples can transfer across models to mislead other models with a high probability (Papernot et al., 2017; Liu et al., 2017). How to effectively defense against adversarial attacks is crucial for security-critical computer vision systems, such as autonomous driving.

As a promising approach, adversarial training defends from adversarial perturbations by training a target classifier with adversarial examples. Researchers have found (Goodfellow et al.; Kurakin

---

[*]Corresponding author.

et al., 2016b; Madry et al., 2018) that adversarial training could increase the robustness of neural networks. However, adversarial training often obtains adversarial examples by taking a specific attack technique (e.g., FGSM) into consideration, so the defense targeted such attack and the trained model exhibits weak generalization ability on adversarial examples from other adversaries (Kurakin et al., 2016b). Tramèr et al. (2018) showed that the robustness of adversarial training can be easily circumvented by the attack that combines with random perturbation from other models. Accordingly, for most existing adversarial training methods, there is a risk of overfitting to adversarial examples crafted on the original model with the specific attack.

In this paper, we propose a novel adversarial training method that is able to improve the generalization of adversarial training. From the perspective of domain adaptation (DA) (Torralba & Efros, 2011), there is a big domain gap between the distribution of clean examples and the distribution of adversarial examples in the high-level representation space, even though adversarial perturbations are imperceptible to humans. Liao et al. (2018) showed that adversarial perturbations are progressively amplified along the layer hierarchy of neural networks, which maximizes the distance between the original and adversarial subspace representations. In addition, adversarial training simply injects adversarial examples from a specific attack into the training set, but there is still a large sample space for adversarial examples. Accordingly, training with the classification loss on such a training set will probably lead to overfitting on the adversarial examples from the specific attack. Even though Wong & Kolter (2018) showed that adversarial training with iterative noisy attacks has stronger robustness than the adversarial training with single-step attacks, iterative attacks have a large computational cost and there is no theoretical analysis to justify that the adversarial examples sampled in such way could be sufficiently representative for the adversarial domain.

Our contributions are focused on how to improve the generalization of adversarial training on the simple yet scalable attacks, such as FGSM (Goodfellow et al.). The key idea of our approach is to formulate the learning procedure as a domain adaptation problem with limited number of target domain samples, where target domain denotes adversarial domain. Specifically, we introduce unsupervised as well as supervised domain adaptation into adversarial training to minimize the gap and increase the similarity between the distributions of clean examples and adversarial examples. In this way, the learned models generalize well on adversarial examples from different $\ell_\infty$ bounded attacks. We evaluate our ATDA method on standard benchmark datasets. Empirical results show that despite a small decay of accuracy on clean data, ATDA significantly improves the generalization ability of adversarial training and has the transfer ability to extend to adversarial training on PGD (Madry et al., 2018).

## 2 Background and Related Work

In this section, we introduce some notations and provides a brief overview of the current advanced attack methods, as well as the defense methods based on adversarial training.

### 2.1 Notation

Denote the clean data domain and the adversarial data domain by $\mathcal{D}$ and $\mathcal{A}$ respectively, we consider a classifier based on a neural network $f(x) : \mathbb{R}^d \to \mathbb{R}^k$. $f(x)$ outputs the probability distribution for an input $x \in [0, 1]^d$, and $k$ denotes the number of classes in the classification task. Let $\varphi$ be the mapping at the logits layer (the last neural layer before the final softmax function), so that $f(x) = softmax(\varphi(x))$. Let $\epsilon$ be the magnitude of the perturbation. Let $x^{adv}$ be the adversarial image computed by perturbing the original image $x$. The cost function of image classification is denoted as $J(x, y)$. We define the *logits* as the logits layer representation, and define the *logit space* as the semantic space of the logits layer representation.

We divide attacks into two types: white-box attacks have the complete knowledge of the target model and can fully access the model; black-box attacks have limited knowledge of the target classifier (e.g.,its architecture) but can not access the model weights.

### 2.2 Attack Methods

We consider four attack methods to generate adversarial examples. For all attacks, the components of adversarial examples are clipped in $[0, 1]$.

**Fast Gradient Sign Method (FGSM).** Goodfellow et al. introduced FGSM to generate adversarial examples by applying perturbations in the direction of the gradient.

$$x^{adv} = x + \epsilon \cdot \text{sign}(\nabla_x J(x, y_{true})) \tag{1}$$

As compared with other attack methods, FGSM is a simple, yet fast and efficient adversary. Accordingly, FGSM is particularly amenable to adversarial training.

**Projected Gradient Descent (PGD).** The Projected Gradient Descent (PGD) adversary was introduced by Madry et al. (2018) without random start, which is a stronger iterative variant of FGSM. This method applies FGSM iteratively for $k$ times with a budget $\alpha$ instead of a single step.

$$
\begin{aligned}
x^{adv_0} &= x \\
x^{adv_{t+1}} &= x^{adv_t} + \alpha \cdot \text{sign}(\nabla_x J(x^{adv_t}, y_{true})) \\
x^{adv_{t+1}} &= \textbf{clip}(x^{adv_{t+1}}, x^{adv_{t+1}} - \epsilon, x^{adv_{t+1}} + \epsilon) \\
x^{adv} &= x^{adv_k}
\end{aligned}
\tag{2}
$$

Here **clip**$(\cdot, a, b)$ function forces its input to reside in the range of $[a, b]$. PGD usually yields a higher success rate than FGSM does in the white-box setting but shows weaker capability in the black-box setting.

**RAND+FGSM (R+FGSM).** Tramèr et al. (2018) proposed R+FGSM against adversarially trained models by applying a small random perturbation of step size $\alpha$ before applying FGSM.

$$
\begin{aligned}
x' &= x + \alpha \cdot \text{sign}(\mathcal{N}(\mathbf{0}^d, \mathbf{I}^d)) \\
x^{adv} &= x' + (\epsilon - \alpha) \cdot \text{sign}(\nabla_x J(x^{adv}, y_{true}))
\end{aligned}
\tag{3}
$$

**Momentum Iterative Method (MIM).** MIM (Dong et al., 2018) is a modification of the iterative FGSM and it won the first place of NIPS 2017 Adversarial Attacks Competition. Its basic idea is to utilize the gradients of the previous $t$ steps with a decay factor $\mu$ to update the gradient at step $t+1$ before applying FGSM with a budget $\alpha$.

$$
\begin{aligned}
x^{adv_0} &= x, \ g_0 = 0 \\
g_{t+1} &= \mu \cdot g_t + \frac{\nabla_x J(x^{adv_t}, y_{true})}{\|\nabla_x J(x^{adv_t}, y_{true})\|_1} \\
x^{adv_{t+1}} &= x^{adv_t} + \alpha \cdot \text{sign}(g_{t+1}) \\
x^{adv_{t+1}} &= \textbf{clip}(x^{adv_{t+1}}, x^{adv_{t+1}} - \epsilon, x^{adv_{t+1}} + \epsilon) \\
x^{adv} &= x^{adv_k}
\end{aligned}
\tag{4}
$$

## 2.3 PROGRESS ON ADVERSARIAL TRAINING

An intuitive technique to defend a deep model against adversarial examples is adversarial training, which injects adversarial examples into the training data during the training process. First, Goodfellow et al. proposed to increase the robustness by feeding the model with both original and adversarial examples generated by FGSM and by learning with the modified objective function.

$$\hat{J}(x, y_{true}) = \alpha J(x, y_{true}) + (1 - \alpha)J(x + \epsilon \cdot \text{sign}(\nabla_x J(x, y_{true})), y_{true}) \tag{5}$$

Kurakin et al. (2016b) scaled the adversarial training to ImageNet (Russakovsky et al., 2015) and showed better results by replacing half the clean example at each batch with the corresponding adversarial examples. Meanwhile, Kurakin et al. (2016b) discovered the *label leaking* effect and suggested not to use the FGSM defined with respect to the true label $y_{true}$. However, their approach has weak robustness to the RAND+FGSM adversary. Tramèr et al. (2018) proposed an *ensemble adversarial training* to improve robustness on black-box attacks by injecting adversarial examples transferred from a number of fixed pre-trained models into the training data.

For adversarial training, another approach is to train only with adversarial examples. Nøkland (2015) proposed a specialization of the method (Goodfellow et al.) that learned only with the objective function of adversarial examples. Madry et al. (2018) demonstrated successful defenses based on adversarial training with the noisy PGD, which randomly initialize an adversarial example within the allowed norm ball before running iterative attack. However, this technique is difficult to scale to large-scale neural networks (Kurakin et al., 2016a) as the iterative attack increases the training time

by a factor that is roughly equal to the number of iterative steps. Wong & Kolter (2018) developed a robust training method by linear programming that minimized the loss for the worst case within the perturbation ball around each clean data point. However, their approach achieved high test error on clean data and it is still challenging to scale to deep or wide neural networks.

As described above, though adversarial training is promising, it is difficult to select a representative adversary to train on and most existing methods are weak in generalization for various adversaries, as the region of the adversarial examples for each clean data is large and contiguous (Tramèr et al., 2017; Tabacof & Valle, 2016). Furthermore, generating a representative set of adversarial examples for large-scale datasets is computationally expensive.

## 3 ADVERSARIAL TRAINING WITH DOMAIN ADAPTATION

In this work, instead of focusing on a better sampling strategy to obtain representative adversarial data from the adversarial domain, we are especially concerned with the problem of how to train with clean data and adversarial examples from the efficient FGSM, so that the adversarially trained model is strong in generalization for different adversaries and has a low computational cost during the training.

We propose an Adversarial Training with Domain Adaptation (ATDA) method to defense adversarial attacks and expect the learned models generalize well for various adversarial examples. Our motivation is to treat the adversarial training on FGSM as a domain adaptation task with limited number of target domain samples, where the target domain denotes adversarial domain. We combine standard adversarial training with the domain adaptor, which minimizes the domain gap between clean examples and adversarial examples. In this way, our adversarially trained model is effective on adversarial examples crafted by FGSM but also shows great generalization on other adversaries.

### 3.1 DOMAIN ADAPTATION ON LOGIT SPACE

#### 3.1.1 UNSUPERVISED DOMAIN ADAPTATION

Suppose we are given some clean training examples $\{x_i\}$ ($x_i \in \mathbb{R}^d$) with labels $\{y_i\}$ from the clean data domain $\mathcal{D}$, and adversarial examples $\{x_i^{adv}\}$ ($x_i^{adv} \in \mathbb{R}^d$) from adversarial data domain $\mathcal{A}$. The adversarial examples are obtained by sampling $(x_i, y_{true})$ from $\mathcal{D}$, computing small perturbations on $x_i$ to generate adversarial perturbations, and outputting $(x_i^{adv}, y_{true})$.

It's known that there is a huge shift in the distributions of clean data and adversarial data in the high-level representation space. Assume that in the *logit space*, data from either the clean domain or the adversarial domain follow a multivariate normal distribution, i.e., $\mathcal{D} \sim \mathcal{N}(\mu_{\mathcal{D}}, \Sigma_{\mathcal{D}})$, $\mathcal{A} \sim \mathcal{N}(\mu_{\mathcal{A}}, \Sigma_{\mathcal{A}})$. Our goal is to learn the logits representation that minimizes the shift by aligning the covariance matrices and the mean vectors of the clean distribution and the adversarial distribution.

To implement the CORrelation ALignment (CORAL), we define a covariance distance between the clean data and the adversarial data as follows.

$$\mathcal{L}_{CORAL}(\mathcal{D}, \mathcal{A}) = \frac{1}{k^2} \left\| C_{\varphi(\mathcal{D})} - C_{\varphi(\mathcal{A})} \right\|_{\ell_1} \tag{6}$$

where $C_{\varphi(\mathcal{D})}$ and $C_{\varphi(\mathcal{A})}$ are the covariance matrices of the clean data and the adversarial data in the logit space respectively, and $\|\cdot\|_{\ell_1}$ denotes the $L_1$ norm of a matrix. Note that $\mathcal{L}_{CORAL}(\mathcal{D}, \mathcal{A})$ is slightly different from the CORAL loss proposed by Sun & Saenko (2016).

Similarly, we use the standard distribution distance metric, Maximum Mean Discrepancy (MMD) (Borgwardt et al., 2006), to minimize the distance of the mean vectors of the clean data and the adversarial data.

$$\mathcal{L}_{MMD}(\mathcal{D}, \mathcal{A}) = \frac{1}{k} \left\| \frac{1}{|\mathcal{D}|} \sum_{x \in \mathcal{D}} \varphi(x) - \frac{1}{|\mathcal{A}|} \sum_{x^{adv} \in \mathcal{A}} \varphi(x^{adv}) \right\|_1 \tag{7}$$

The loss function for Unsupervised Domain Adaptation (UDA) can be calculated as follows.

$$\mathcal{L}_{UDA}(\mathcal{D}, \mathcal{A}) = \mathcal{L}_{CORAL}(\mathcal{D}, \mathcal{A}) + \mathcal{L}_{MMD}(\mathcal{D}, \mathcal{A}) \tag{8}$$

### 3.1.2 SUPERVISED DOMAIN ADAPTATION

Even though the unsupervised domain adaptation achieves perfect confusion alignment, there is no guarantee that samples of the same label from clean domain and adversarial domain would map nearby in the logit space. To effectively utilize the labeled data in the adversarial domain, we introduce a supervised domain adaptation (SDA) by proposing a new loss function, denoted as margin loss, to minimize the intra-class variations and maximize the inter-class variations on samples of different domains. The SDA loss is shown in Eq. (9).

$$
\begin{aligned}
\mathcal{L}_{SDA}(\mathcal{D},\mathcal{A}) =& \mathcal{L}_{margin}(\mathcal{D},\mathcal{A}) \\
=& \frac{1}{(k-1)(|\mathcal{D}|+|\mathcal{A}|)} \cdot \\
& \sum_{x \in \mathcal{D} \cup \mathcal{A}} \sum_{c^n \in C \backslash \{c_{y_{true}}\}} softplus(\|\varphi(x) - c_{y_{true}}\|_1 - \|\varphi(x) - c^n\|_1)
\end{aligned}
\tag{9}
$$

Here $softplus$ denotes a function $ln(1 + exp(\cdot))$; $c_{y_{true}} \in \mathbb{R}^k$ denotes the center of $y_{true}$ class in the logit space; $C = \{ c_j \mid j = 1, 2, ..., k \}$ is a set consisting of the logits center for each class, which will be updated as the logits changed. Similar to the center loss (Wen et al., 2016), we update center $c_j$ for each class $j$:

$$
\Delta c_j^t = \frac{\sum_{x \in \mathcal{D} \cup \mathcal{A}} \mathbf{1}_{y_{true}=j} \cdot (c_j^t - \varphi(x))}{1 + \sum_{x \in \mathcal{D} \cup \mathcal{A}} \mathbf{1}_{y_{true}=j}}
$$
$$
c_j^{t+1} = c_j^t - \alpha \cdot \Delta c_j^t
\tag{10}
$$

where $\mathbf{1}_{condition} = 1$ if the *condition* is true, otherwise $\mathbf{1}_{condition} = 0$; $\alpha$ denotes the learning rate of the centers. During the training process, the logits center for each class can integrate the logits representation from both the clean domain and the adversarial domain.

## 3.2 ADVERSARIAL TRAINING

For adversarial training, iterative attacks are fairly expensive to compute and single-step attacks are fast to compute. Accordingly, we use a variant of FGSM attack (Kurakin et al., 2016b) that avoids the *label leaking* effect to generate a new adversarial example $x_i^{adv}$ for each clean example $x_i$.

$$
x_i^{adv} = x_i + \epsilon \cdot \text{sign}(\nabla_x J(x_i, y_{target}))
\tag{11}
$$

where $y_{target}$ denotes the predicted class $\arg\max\{\varphi(x_i)\}$ of the model.

However, in this case, the sampled adversarial examples are aggressive but not sufficiently representative due to the fact that the sampled adversarial examples always lie at the boundary of the $\ell_\infty$ ball of radius $\epsilon$ (see Figure 1) and the adversarial examples within the boundary are ignored. For adversarial training, if we train a deep neural network only on the clean data and the adversarial data from the FGSM attack, the adversarially trained model will overfit on these two kinds of data and exhibits weak generalization ability on the adversarial examples sampled from other attacks. From a different perspective, such problem can be viewed as a domain adaptation problem with limited number of labeled target domain samples, as only some special data point can be sampled in the adversarial domain by FGSM adversary.

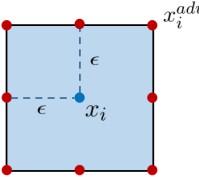

Figure 1: Illustration of the adversarial sampling by FGSM for $x_i \in \mathbb{R}^2$. The blue dot (in the center) represents a clean example and the red dots (along the boundary) represent the potential adversarial examples for the clean example.

Consequently, it is natural to combine the adversarial training with domain adaptation to improve the generalization ability on adversarial data. We generate new adversarial examples by the variant of FGSM attack shown in Eq. (11), then we use the following loss function to meet the criteria of domain adaptation while training a strong classifier.

$$\begin{aligned}
\mathcal{L}(\mathcal{D}, \mathcal{A}) &= \mathcal{L}_C(\mathcal{D}) + \mathcal{L}_C(\mathcal{A}) + \lambda \cdot \mathcal{L}_{DA}(\mathcal{D}, \mathcal{A}) \\
&= \mathcal{L}_C(\mathcal{D}) + \mathcal{L}_C(\mathcal{A}) + \lambda \cdot (\mathcal{L}_{UDA}(\mathcal{D}, \mathcal{A}) + \mathcal{L}_{SDA}(\mathcal{D}, \mathcal{A})) \\
&= \frac{1}{m} \sum_{x \in \mathcal{D}} \mathcal{L}_C(x|y_{true}) + \frac{1}{m} \sum_{x^{adv} \in \mathcal{A}} \mathcal{L}_C(x^{adv}|y_{true}) \\
&\quad + \lambda \cdot (\mathcal{L}_{CORAL}(\mathcal{D}, \mathcal{A}) + \mathcal{L}_{MMD}(\mathcal{D}, \mathcal{A}) + \mathcal{L}_{margin}(\mathcal{D}, \mathcal{A}))
\end{aligned} \tag{12}$$

Here $\lambda$ is the hyper-parameter to balance the regularization term; $m$ is the number of input clean examples; $\mathcal{D}$ indicates the input clean examples $\{x_i\}$, and $\mathcal{A}$ the corresponding adversarial examples $\{x_i^{adv}\}$; $\mathcal{L}_C$ denotes the classification loss. The training process is summarized in Algorithm 1.

---

**Algorithm 1** Adversarial training with domain adaptation on network $f(x) : \mathbb{R}^d \to \mathbb{R}^k$.
Parameters: Size of the training minibatch is $m$.

---

1: Randomly initialize network $f(x)$ and logits centers $\{c_j \mid j = 1, 2, ..., k\}$;
2: Number of iterations $t \leftarrow 0$;
3: **repeat**
4:     $t \leftarrow t + 1$;
5:     Read a minibatch of data $\mathcal{D}_b = \{x_1, ..., x_m\}$ from the training set;
6:     Use the current state of network $f$ to generate adversarial examples $\mathcal{A}_b = \{x_1^{adv}, ..., x_m^{adv}\}$ by the FGSM variant that avoids label leaking;
7:     Extract logits for examples $\mathcal{D}_b$, $\mathcal{A}_b$ by performing forward-backward propagation from the input layer to the logits layer $\varphi(x)$;
8:     Update parameters $c_j$ for each class $j$ by $c_j^{t+1} = c_j^t - \alpha \cdot \Delta c_j^t$;
9:     Compute the loss by Eq. (12) and update parameters of network $f$ by back propagation;
10: **until** the training converges.

---

## 4 EXPERIMENTS

In this section, we evaluate our ATDA method on various benchmark datasets to demonstrate the robustness and contrast its performance against other competing methods under different white-box and black-box attacks with bounded $\ell_\infty$ norm. Code for these experiments is available at `https://github.com/JHL-HUST/ATDA`.

### 4.1 EXPERIMENTAL SETUP

**Datasets.** We consider four popular datasets, namely Fashion-MNIST (Xiao et al., 2017), SVHN (Netzer et al., 2011), CIFAR-10 and CIFAR-100 (Krizhevsky & Hinton, 2009). For all experiments, we normalize the pixel values to $[0, 1]$ by dividing 255.

**Baselines.** To evaluate the generalization power on adversarial examples in both the white-box and black-box settings, we report the clean test accuracy, the defense accuracy on FGSM, PGD, R+FGSM and MIM in the non-targeted way. The common settings for these attacks are shown in Table 5 of the Appendix. We compare our ATDA method with normal training as well as several state-of-the-art adversarial training methods:

- Normal Training (NT). Training with cross-entropy loss on the clean training data.
- Standard Adversarial Training (SAT) (Goodfellow et al.). Training with the cross-entropy on the clean training data and the adversarial examples from the FGSM variant with perturbation $\epsilon$ to avoid label leaking.
- Ensemble Adversarial Training (EAT) (Tramèr et al., 2018). Training with cross-entropy on the clean training data and the adversarial examples crafted from the currently trained model and the static pre-trained models by the FGSM variant with the perturbation $\epsilon$ to avoid label leaking.
- Provably Robust Training (PRT) (Wong & Kolter, 2018). Training with cross-entropy loss on the worst case in the $\ell_\infty$ ball of radius $\epsilon$ around each clean training data point. It could be seen as training with a complicated method of sampling in the $\ell_\infty$ ball of radius $\epsilon$.

**Evaluation Setup.** For each benchmark dataset, we train a normal model and various adversarial models with perturbation $\epsilon$ on a main model with ConvNet architecture, and evaluate them on

various attacks bounded by $\epsilon$. Moreover, for Ensemble Adversarial Training (EAT), we use two different models as the static pre-trained models. For black-box attacks, we test trained models on the adversarial examples transferred from a model held out during the training. All experiments are implemented on a single Titan X GPU. For all experiments, we set the hyper-parameter $\lambda$ in Eq. (12) to $1/3$ and the hyper-parameter $\alpha$ in Eq. (10) to $0.1$. For more details about neural network architectures and training hyper-parameters, see Appendix A. We tune the networks to make sure they work, not to post concentrates on optimizing these settings.

## 4.2 Comparison of Defense Performance on Accuracy

We evaluate the defense performance of our ATDA method from the perspective of classification accuracy on various datasets, and compare with the baselines.

**Evaluation on Fashion-MNIST.** The accuracy results on Fashion-MNIST are reported in Table 1a. NT yields the best performance on the clean data, but generalizes poorly on adversarial examples. SAT and EAT overfit on the clean data and the adversarial data from FGSM. PRT achieves lower error against various adversaries, but higher error on the clean data. ATDA achieves stronger robustness against different $\ell_\infty$ bounded adversaries as compared to SAT (adversarial training on FGSM).

**Evaluation on SVHN.** The classification accuracy on SVHN are summarized in Table 1b. PRT seems to degrade the performance on the clean testing data and exhibits weak robustness on various attacks. As compared to SAT, ATDA achieves stronger generalization ability on adversarial examples from various attacks and higher accuracy on the white-box adversaries, at the same time it only loses a negligible performance on clean data.

**Evaluation on CIFAR-10.** Compared with Fashion-MNIST and SVHN, CIFAR-10 is a more difficult dataset for classification. As PRT is challenging and expensive to scale to large neural networks due to its complexity, the results of PRT are not reported. The accuracy results on CIFAR-10 are summarized in Table 1c. ATDA outperforms all the competing methods on most adversaries, despite a slightly lower performance on clean data.

**Evaluation on CIFAR-100.** The CIFAR-100 dataset contains 100 image classes, with 600 images per class. Our goal here is not to achieve state-of-the-art performance on CIFAR-100, but to compare the generalization ability of different training methods on a comparatively large dataset. The results on CIFAR-100 are summarized in Table 1d. Compared to SAT, ATDA achieves better generalization on various adversarial examples and it does not degrade the performance on clean data.

In conclusion, the accuracy results provide empirical evidence that ATDA has great generalization ability on different adversaries as compared to SAT and outperforms other competing methods.

## 4.3 Further Analysis on the Defense Performance

To further investigate the defence performance of the proposed method, we compute two other metrics: the local loss sensitivity to perturbations and the shift of adversarial data distribution with respect to the clean data distribution.

**Local Loss Sensitivity.** One method to quantify smoothness and generalization to perturbations for models is the local loss sensitivity (Arpit et al., 2017). It is calculated in the clean testing data as follows. The lower the value is, the smoother the loss function is.

$$\mathcal{S} = \frac{1}{m} \sum_{i=1}^{m} \|\nabla_x J(x_i, y_i)\|_2 \tag{13}$$

The results of the local loss sensitivity for the aforementioned learned models are summarized in Table 2. The results suggest that adversarial training methods do increase the smoothness of the model as compared with the normal training and ATDA performs the best.

**Distribution Discrepancy.** To quantify the dissimilarity of the distributions between the clean data and the adversarial data, we compare our learned logits embeddings with the logits embeddings of the competing methods on Fashion-MNIST. We use t-SNE (Maaten & Hinton, 2008) for the comparison on the training data, testing data and adversarial testing data from the white-box FGSM or PGD. The comparisons are illustrated in Figure 2 and we report the detailed MMD distances across domains in Table 3. Compared with NT, SAT and EAT actually increase the MMD distance

across domains of the clean data and the adversarial data. In contrast, PRT and ATDA can learn domain invariance between the clean domain and the adversarial domain. Furthermore, our learned logits representation achieves the best performance on domain invariance.

Table 1: **The accuracy of defense methods on the testing datasets and the adversarial examples generated by various adversaries.**

(a) **On Fashion-MNIST.** The magnitude of perturbations is 0.1 in $\ell_\infty$ norm.

| Defense | Clean (%) | White-Box Attack (%) | | | | Black-Box Attack (%) | | | |
|---|---|---|---|---|---|---|---|---|---|
| | | FGSM | PGD | R+FGSM | MIM | FGSM | PGD | R+FGSM | MIM |
| NT | 90.5 | 8.3 | 0.1 | 15.0 | 0.1 | 52.1 | 52.6 | 68.2 | 44.3 |
| SAT | **90.9** | 88.8 | 7.4 | 31.2 | 9.4 | 79.8 | 80.1 | 81.8 | 80.0 |
| EAT | 90.8 | **89.0** | 4.3 | 31.6 | 6.6 | 80.8 | 81.4 | 82.3 | 78.8 |
| PRT | 76.9 | 67.4 | 66.8 | 72.2 | 66.7 | 75.5 | 75.5 | 76.4 | 75.4 |
| ATDA | 85.5 | 78.2 | **68.6** | **77.0** | **68.8** | **83.8** | **83.7** | **84.5** | **83.3** |

(b) **On SVHN.** The magnitude of perturbations is 0.02 in $\ell_\infty$ norm.

| Defense | Clean (%) | White-Box Attack (%) | | | | Black-Box Attack (%) | | | |
|---|---|---|---|---|---|---|---|---|---|
| | | FGSM | PGD | R+FGSM | MIM | FGSM | PGD | R+FGSM | MIM |
| NT | 84.9 | 19.6 | 3.6 | 33.3 | 4.6 | 64.3 | 68.0 | 76.5 | 64.8 |
| SAT | 86.6 | 52.1 | 44.4 | 70.4 | 46.1 | 79.0 | 79.7 | 83.3 | 78.7 |
| EAT | **88.6** | 47.1 | 34.4 | 67.6 | 36.6 | **80.4** | **81.3** | **85.3** | **80.2** |
| PRT | 58.5 | 41.7 | 41.1 | 49.5 | 41.2 | 53.7 | 54.9 | 56.1 | 54.0 |
| ATDA | 82.9 | **57.2** | **53.2** | **70.6** | **53.9** | 75.3 | 76.4 | 79.5 | 75.4 |

(c) **On CIFAR-10.** The magnitude of perturbations is 4/255 in $\ell_\infty$ norm.

| Defense | Clean (%) | White-Box Attack (%) | | | | Black-Box Attack (%) | | | |
|---|---|---|---|---|---|---|---|---|---|
| | | FGSM | PGD | R+FGSM | MIM | FGSM | PGD | R+FGSM | MIM |
| NT | **86.9** | 4.3 | 0.5 | 19.6 | 0.9 | 41.3 | 23.8 | 60.9 | 25.8 |
| SAT | 86.2 | 52.4 | 49.5 | 70.2 | 50.5 | 80.5 | 80.5 | **83.5** | 80.3 |
| EAT | 86.0 | 46.7 | 43.5 | 67.5 | 44.5 | 80.0 | 80.1 | 83.2 | 79.8 |
| PRT | - | - | - | - | - | - | - | - | - |
| ATDA | 84.8 | **60.7** | **58.1** | **73.2** | **59.0** | **80.7** | **80.7** | 83.0 | **80.6** |

(d) **On CIFAR-100.** The magnitude of perturbations is 4/255 in $\ell_\infty$ norm.

| Defense | Clean (%) | White-Box Attack (%) | | | | Black-Box Attack (%) | | | |
|---|---|---|---|---|---|---|---|---|---|
| | | FGSM | PGD | R+FGSM | MIM | FGSM | PGD | R+FGSM | MIM |
| NT | 59.0 | 0.2 | 0.4 | 6.1 | 0.4 | 28.8 | 23.8 | 41.6 | 24.2 |
| SAT | 58.7 | 17.7 | 18.0 | 34.8 | 17.9 | 53.2 | 53.1 | 55.8 | 53.0 |
| EAT | 59.1 | 12.5 | 13.5 | 31.1 | 13.2 | 52.0 | 52.2 | 55.7 | 51.9 |
| PRT | - | - | - | - | - | - | - | - | - |
| ATDA | **61.6** | **29.3** | **26.2** | **43.0** | **27.3** | **56.0** | **56.0** | **58.7** | **56.0** |

Table 2: **The local loss sensitivity analysis for defense methods.**

| Dataset | Local loss sensitivity | | | | |
|---|---|---|---|---|---|
| | NT | SAT | EAT | PRT | ATDA |
| Fashion-MNIST | 5.84 | 5.52 | 3.24 | 0.56 | **0.49** |
| SVHN | 13.48 | 2.03 | 2.41 | 1.79 | **1.64** |
| CIFAR-10 | 6.56 | 1.61 | 2.08 | - | **0.90** |
| CIFAR-100 | 23.16 | 7.13 | 8.40 | - | **2.67** |

## 4.4 ABLATION STUDIES ON ATDA

To individually dissect the effectiveness of different components in ATDA (Standard Adversarial Training (SAT), Unsupervised Domain Adaptation (UDA), and Supervised Domain Adaptation (SDA)), we conduct a series of ablation experiments in Figure 3. For each model, we report the

Table 3: **The MMD distance across domains in the logit space for defense methods on Fashion-MNIST.** $\mathcal{D}$ denotes the distribution of the clean testing data; $\mathcal{A}_{FGSM}$ and $\mathcal{A}_{PGD}$ denote the distributions of the adversarial testing data generated by the white-box FGSM and PGD, respectively.

| MMD Distance | Defense Method | | | | |
|---|---|---|---|---|---|
| | NT | SAT | EAT | PRT | ATDA |
| $\mathrm{MMD}(\mathcal{D}, \mathcal{A}_{FGSM})$ | 2.174 | 5.353 | 5.120 | 0.098 | **0.005** |
| $\mathrm{MMD}(\mathcal{D}, \mathcal{A}_{PGD})$ | 5.287 | 2.909 | 1.239 | 0.101 | **0.019** |

(a) Training data  (b) Testing data  (c) FGSM adversarial data  (d) PGD adversarial data

Figure 2: t-SNE visualizations for the embeddings of training data, testing data, and adversarial testing data from FGSM and PGD in the logit space for Fashion-MNIST. The first row to the fifth row correspond to NT, SAT, EAT, PRT and ATDA, respectively.

*average* accuracy rates over all white-box attacks and all black-box attacks, respectively. The results illustrate that, by aligning the covariance matrix and mean vector of the clean and adversarial examples, UDA plays a key role in improving the generalization of SAT on various attacks. In general, the aware of margin loss on SDA can also improve the defense quality on standard adversarial training, but the effectiveness is not very stable over all datasets. By combining UDA and SDA together with SAT, our final algorithm ATDA can exhibits stable improvements on the standard adversarial training. In general, the performance of ATDA is slightly better than SAT+UDA.

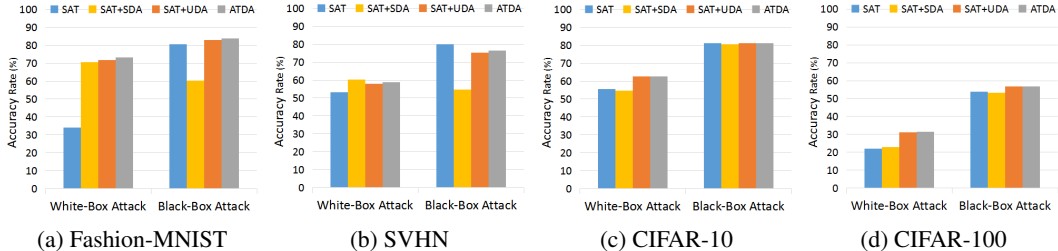

Figure 3: Ablation experiments for ATDA to investigate the impact of Standard Adversarial Training (SAT), Unsupervised Domain Adaptation (UDA), and Supervised Domain Adaptation (SDA). We report the *average* accuracy rates over all white-box attacks and all black-box attacks, respectively.

## 4.5 Extension to PGD-Adversarial Training

ATDA can simply be extended to adversarial training on other adversaries. We now consider to extend the ATDA method to PGD-Adversarial Training (PAT) (Madry et al., 2018): adversarial training on the noisy PGD with perturbation $\epsilon$. By combining adversarial training on the noisy PGD with domain adaptation, we implement an extension of ATDA for PAT, called PATDA. For the noisy PGD, we set the iterated step $k$ as 10 and the budget $\alpha$ as $\epsilon/4$ according to Madry et al. (2018).

As shown in Table 4, we evaluate the defense performance of PAT and PATDA on various datasets. On Fashion-MNIST, we observe that PATDA fails to increase robustness to most adversaries as compared to PAT. On SVHN, PAT and PATDA fail to converge properly. The results are not surprising, as training with the hard and sufficient adversarial examples (from the noisy PGD) requires the neural networks with more parameters. On CIFAR-10 and CIFAR-100, PATDA achieves stronger robustness to various attacks than PAT. In general, PATDA exhibits stronger robustness to various adversaries as compared to PAT. The results indicate that domain adaptation can be applied flexibly to adversarial training on other adversaries to improve the defense performance.

Table 4: **The accuracy of PAT and PATDA on the testing datasets and the adversarial examples generated by various adversaries.** The magnitude of perturbations in $\ell_\infty$ norm is 0.1 for Fashion-MNIST, 0.02 for SVHN, and $4/255$ for CIFAR-10 and CIFAR-100.

| Dataset | Defense | Clean (%) | White-Box Attack (%) | | | | Black-Box Attack (%) | | | |
|---|---|---|---|---|---|---|---|---|---|---|
| | | | FGSM | PGD | R+FGSM | MIM | FGSM | PGD | R+FGSM | MIM |
| Fashion-MNIST | PAT | **85.3** | **78.7** | **76.5** | **81.7** | **76.7** | 81.8 | **83.9** | **84.8** | **83.8** |
| | PATDA | 83.2 | 77.0 | 75.5 | 79.8 | 75.7 | **84.0** | 81.7 | 82.4 | 81.6 |
| SVHN | PAT | 19.6 | 19.6 | 19.6 | 19.6 | 19.6 | 19.6 | 19.6 | 19.6 | 19.6 |
| | PATDA | 19.6 | 19.6 | 19.6 | 19.6 | 19.6 | 19.6 | 19.6 | 19.6 | 19.6 |
| CIFAR-10 | PAT | **83.4** | 55.1 | 53.0 | 70.2 | 53.8 | **79.9** | 80.0 | **81.8** | **79.9** |
| | PATDA | **83.4** | **62.2** | **60.2** | **73.4** | **61.0** | **79.9** | **80.1** | 81.7 | **79.9** |
| CIFAR-100 | PAT | 55.4 | 26.2 | 24.0 | 38.9 | 24.9 | 52.4 | 52.3 | 54.1 | 52.3 |
| | PATDA | **59.4** | **32.5** | **30.9** | **44.9** | **31.5** | **55.3** | **55.2** | **57.4** | **55.1** |

## 5 Conclusion

In this study, we regard the adversarial training as a domain adaptation task with limited number of target labeled data. By combining adversarial training on FGSM adversary with unsupervised and supervised domain adaptation, the generalization ability on adversarial examples from various attacks and the smoothness on the learned models can be highly improved for robust defense. In addition, ATDA can easily be extended to adversarial training on iterative attacks (e.g., PGD) to improve the defense performance. The experimental results on several benchmark datasets suggest that the proposed ATDA and its extension PATDA achieve significantly better generalization results as compared with current competing adversarial training methods.

## Acknowledgments

This work is supported by National Natural Science Foundation (61772219).

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

## A  EXPERIMENTAL DETAILS

In the appendix, we show all details of the common settings, neural network architectures and training hyper-parameters for the experiments.

### A.1  HYPER-PARAMETERS FOR ADVERSARIES.

For each dataset, the details about the hyper-parameters of various adversaries are shown in Table 5, where $\epsilon$ denotes the magnitude of adversarial perturbations.

Table 5: **Common settings of attacks for all experiments**

| Attack | Parameter | Norm |
|---|---|---|
| FGSM | N/A | $\ell_\infty$ |
| PGD | Iterated step $k = 20$, $\alpha = \epsilon/10$ | $\ell_\infty$ |
| R+FGSM | Random perturbation $\alpha = \epsilon/2$ | $\ell_\infty$ |
| MIM | Iterated step $k = 10$, $\alpha = \epsilon/5$, $\mu = 1.0$ | $\ell_\infty$ |

### A.2  NEURAL NETWORK ARCHITECTURES AND TRAINING HYPER-PARAMETERS

**Fashion-MNIST.**  In the training phase, we use Adam optimizer with a learning rate of 0.001 and set the batch size to 64. For Fashion-MNIST, the neural network architectures for the main model, the static pre-trained models and the model held out during training are depicted in Table 6. For all adversarial training methods, the magnitude of perturbations is 0.1 in $\ell_\infty$ norm.

**SVHN.**  In the training phase, we use Adam optimizer with a learning rate of 0.001 and set the batch size to 32 and use the same architectures as in Fashion-MNIST. For all adversarial training methods, the magnitude of perturbations is 0.02 in $\ell_\infty$ norm.

Table 6: **Neural network architectures used for the Fashion-MNIST and SVHN datasets**. Conv: convolutional layer with Relu, FC: fully connected layer.

| Main model | Pre-trained $\text{model}_A$ | Pre-trained $\text{model}_A$ | Holdout model |
|---|---|---|---|
| Conv(16, 4x4) | Conv(32, 5x5) | Dropout(0.2) | Conv(64, 3x3) |
| Conv(32, 4x4) | Conv(32, 5x5) | Conv(32, 3x3) | FC(300) + Relu |
| FC(100) + Relu | Dropout(0.1) | Conv(32, 3x3) | Dropout(0.5) |
| FC(10) | FC(128) + Relu | FC(128) + Relu | FC(300) + Relu |
|  | Dropout(0.5) | Dropout(0.5) | Dropout(0.5) |
|  | FC(10) | FC(10) | FC(10) |

**CIFAR-10.**  In the training phase, we use the same training settings as in SVHN. we use Adam optimizer with a learning rate of $0.001$ and set the batch size to 32. In order to enhance the expressive power of deep neural networks, we use Exponential Linear Unit (ELU) (Clevert et al., 2015) as the activation function and introduce Group Normalization (Wu & He, 2018) into the architectures. The neural network architectures for CIFAR-10 are shown in Table 7. For all adversarial training methods, the magnitude of perturbations is $4/255$ in $\ell_\infty$ norm.

**CIFAR-100.**  We use the same training settings as in CIFAR-10. For CIFAR-100, the neural network architectures for the main model, the static pre-trained models and the model held out during training are shown in Table 8. For all adversarial training methods, the magnitude of perturbations is $4/255$ in $\ell_\infty$ norm.

Table 7: **Neural network architectures used for the CIFAR-10 dataset**. Conv: convolutional layer with Group Normalization and ELU; GAP: global average pooling.

| Main model | Pre-trained $\text{model}_A$ | Pre-trained $\text{model}_B$ | Holdout model |
|---|---|---|---|
| Conv(96, 3x3) | Conv(96, 5x5) | Conv(192, 5x5) | Conv(96, 3x3) |
| Conv(96, 3x3) | Conv(96, 1x1) | Conv(96, 1x1) | Dropout(0.2) |
| Conv(96, 3x3) | MaxPooling(3x3, 2) | MaxPooling(3x3, 2) | Conv(96, 3x3) x 2 |
| Dropout(0.5) | Dropout(0.5) | Dropout(0.5) | Dropout(0.5) |
| Conv(192, 3x3) x 3 | Conv(192, 5x5) | Conv(192, 5x5) | Conv(192, 3x3) x 2 |
| Dropout(0.5) | Conv(192, 1x1) | Conv(192, 1x1) | Dropout(0.5) |
| Conv(192, 3x3) | MaxPooling(3x3, 2) | MaxPooling(3x3, 2) | Conv(256, 3x3) |
| Conv(192, 1x1) | Dropout(0.5) | Dropout(0.5) | Conv(256, 1x1) |
| Conv(10, 1x1) | Conv(256, 3x3) | Conv(256, 3x3) | Conv(10, 1x1) |
| GAP | Conv(256, 1x1) | Conv(256, 1x1) | GAP |
| | Conv(10, 1x1) | Conv(10, 1x1) | |
| | GAP | GAP | |

Table 8: **Neural network architectures used for the CIFAR-100 dataset**. Conv: convolutional layer with Group Normalization and ELU; GAP: global average pooling.

| Main model | Pre-trained $\text{model}_A$ | Pre-trained $\text{model}_B$ | Holdout model |
|---|---|---|---|
| Conv(96, 3x3) | Conv(96, 5x5) | Conv(192, 5x5) | Conv(96, 3x3) |
| Conv(96, 3x3) | Conv(96, 1x1) | Conv(96, 1x1) | Dropout(0.2) |
| Conv(96, 3x3) | MaxPooling(3x3, 2) | MaxPooling(3x3, 2) | Conv(96, 3x3) x 2 |
| Dropout(0.5) | Dropout(0.5) | Dropout(0.5) | Dropout(0.5) |
| Conv(192, 3x3) x 3 | Conv(192, 5x5) | Conv(192, 5x5) | Conv(192, 3x3) x 2 |
| Dropout(0.5) | Conv(192, 1x1) | Conv(192, 1x1) | Dropout(0.5) |
| Conv(192, 3x3) | MaxPooling(3x3, 2) | MaxPooling(3x3, 2) | Conv(256, 3x3) |
| Conv(192, 1x1) | Dropout(0.5) | Dropout(0.5) | Conv(256, 1x1) |
| Conv(100, 1x1) | Conv(256, 3x3) | Conv(256, 3x3) | Conv(100, 1x1) |
| GAP | Conv(256, 1x1) | Conv(256, 1x1) | GAP |
| | Conv(100, 1x1) | Conv(100, 1x1) | |
| | GAP | GAP | |

