# OpenReview forum: "Improving the Generalization of Adversarial Training with Domain Adaptation"
_ICLR.cc/2019/Conference_

### Official Review · AnonReviewer3 · 2018-10-31
**Good paper**

**Rating:** 6
**Confidence:** 2

**Review:**

This paper addresses the generalization of adversarial training by proposing a new domain adaptation method. In order to have robust defense for adversarial examples, they combine supervised and unsupervised learning for domain adaptation. The idea of domain adaptation is to increase the similarity between clear and adversarial examples. For this purpose, in their objective, they are minimizing the domain shift by aligning the covariance matrix and mean vector of the clean and adversarial examples.

From experimental viewpoint, they have lower performance than almost all competitors on clean data, but they are beating them when there is white-box as well as the back-box threats. This means their method gives a good generalization. In CIFAR-100 they do not have this trade-off for accuracy and generalization; they are beating other competitors in clean data as well.

The paper is clear and well-written. The introduction and background give useful information.

In general, I think the paper has a potential for acceptance, but I have to mention that I am not an expert in Adversarial networks area.

---

> ### Author Response · Authors · 2018-11-22
> **More experiments and analysis are added. Thank you for your helpful review.**
>
> We deeply appreciate your positive comments. In the revision, we reorganized the content of the Experimental Section, improved the writing quality and organization, and add more quantitative analysis to strengthen our work.
>
> 1) We add subsection 4.3 and evaluate the robustness of defenses in terms of the local loss sensitivity to perturbations and the shift of adversarial data distribution with respect to the clean data distribution. ATDA performs the best in terms of the two metrics.
>
> 2) We add subsection 4.4 and conduct a series of ablation experiments to tease out the benefit of each of the various terms added to the loss functions. Both UDA and SDA can improve the generalization of SAT on various attacks. By combining UDA and SDA together with SAT, the final ATDA can exhibits stable improvements on the standard adversarial training.
>
> 3) We add subsection 4.5 and verify the scalability and flexibility of ATDA by combining adversarial training on the noisy PGD with domain adaptation. The results indicate domain adaptation can be applied flexibly to adversarial training on other adversaries to improve the robustness of the model.
>
> We have tried to address all concerns of the reviewers and changed the abstract, introduction and conclusion accordingly. We have also revised the paper to address the minor typos. We hope our effort can earn your support and convince you the quality of our work.

---

### Official Review · AnonReviewer1 · 2018-11-02
**A new adversarial training with domain adaptation demonstrating fair performance improvement**

**Rating:** 6
**Confidence:** 4

**Review:**

Authors propose a new adversarial training with domain adaptation method to overcome the weak generalisation problem in adversarial training for adversarial examples from different attacks. Authors consider the adversarial training as a domain adaptation task with limited number of target labeled data. They demonstrate that by combining unsupervised and supervised domain adaptation with adversarial training, the generalisation ability on adversarial examples from various attacks can be improved for efficient defence. The experimental results on several benchmark datasets suggest that
the proposed approach achieves significantly better generalisation results in most cases, when compared to current
competing adversarial training methods. Paper is clearly written and well structured. The novelty of the proposed technique is fair and the originality alike. The results are not very conclusive therefore I think more experiments are needed and possible further adjustments.

---

> ### Author Response · Authors · 2018-11-22
> **More experiments and analysis are added. Thank you for the helpful review.**
>
> We deeply appreciate the reviewer for the positive remarks and constructive suggestions. Your suggestion of further experiments is a great idea and towards this, we have performed the following revisions.
>
> 1. We have reorganized the content of the Experimental Section, clarify the writing  and reduce the duplication of expression, so as to leave more space to describe further quantitative analysis on ATDA.
>
> 2. We add subsection 4.3 and performed additional experiments to measure the defense efficiency of ATDA in terms of the local loss sensitivity (defined in [1]) to perturbations and the shift of adversarial data distribution with respect to the clean data distribution.
>     The results in Table 2 on local loss sensitivity suggest that the adversarial training methods do increase the smoothness of the learned model as compared with the normal training, and ATDA performs the best in terms of the sensitivity of the loss function.
>     We also quantify the distribution discrepancy of the clean data and the adversarial data. Besides the illustration of the t-SNE embeddings (Figure 2), we report the detailed MMD distances across domains in Table 3 for quantitative analysis (the lower the better). The results suggest that SAT and EAT actually increase the MMD distance across domains of the clean data and the adversarial data. In contrast, PRT and ATDA can learn domain invariance between the clean domain and the adversarial domain, and ATDA achieves the best performance in terms of the distribution discrepancy.
> [1] https://arxiv.org/abs/1706.05394 (ICML 2017).
>
> 3. To tease out the benefit of each of the various terms added to the loss functions, as Reviewer 2 suggested, we study the benefit of the different components in ATDA:
> 	1) Standard Adversarial Training (SAT)
> 	2) Unsupervised Domain Adaptation (UDA)
> 	3) Supervised Domain Adaptation (SDA)
>     We add subsection 4.4 and conduct a series of ablation experiments. The results are shown in Figure 3. We observe that by aligning the covariance matrix and mean vector of the clean and adversarial examples, UDA plays a key role in improving the generalization of SAT on various attacks.
> 	In general, the aware of margin loss on SDA can also improve the defense quality on standard adversarial training, but the effectiveness is not very stable over all datasets. By combining UDA and SDA together with SAT, our final algorithm ATDA can exhibits stable improvements on the standard adversarial training. In general, the performance is slightly better than SAT+UDA.
>
> 4. Although our work is mainly focused on the adversarial training yet efficient FGSM adversary, we add subsection 4.5 to extend our domain adaptation method to PGD-Adversarial Training (PAT) and get an extension of ATDA (called PATDA).
>    We performed experiments to evaluate the robustness of PAT and PATDA. The results suggest that PATDA has stronger robustness on various attacks with respect to PAT. This indicates domain adaptation can be applied to adversarial training on other adversaries to improve the robustness of the model.
>
> We thank you again for the valuable feedback and comments, which have improved the manuscript apparently. We have addressed all the concerns and changed the abstract, introduction and conclusion accordingly in the revision of the paper. We hope the additional quantitative results can convince you the quality of our work.

---

### Official Review · AnonReviewer2 · 2018-11-06
**Good idea and experimental evidence but lacking in rigor and more empirical analysis**

**Rating:** 6
**Confidence:** 3

**Review:**

The paper casts the problem of learning from adversarial examples to make models resistant to adversarial perturbations to a domain adaptation problem. The proposed method Adversarial training with Domain adapatation( ATDA) learns a representation that is invariant to clean and adversarial data achieving state of the art results on CIFAR.

quality - Paper is well written, explanation of the mathematical parts are good, experimental quality can be much better.
clarity - the problem motivation as well as the methodology is clearly explained. the learning from the experiments are unclear and need more work.
originality - The casting of the problem as domain adaptation is original but from the experiments it was not conclusive as to how much benefit we get.
significance of this work -  Current models being sensitive to adversarial perturbations is quite a big problem so the particular problem authors are trying to address is very significant.

pros

A good idea, enough experiments that indicate the benefit of casting this as a domain adaptation problem.

cons

I feel, the authors should have extended the experiments to ImageNet which is a much larger dataset and validate the findings still hold, I feel the discussion section and comparison to other methods needs to be worked to be more thorough and to tease out the benefit of each of the various terms added to the loss functions as currently all we have is final numbers without much explanation and details. TSNE embeddings part is also very qualitative and while the plots indicate a better separation for ATDA, I feel authors should do more quantitative analysis on the embeddings instead of just qualitative plots.

---

> ### Author Response · Authors · 2018-11-22
> **More experiments and analysis are added. Thank you for your helpful review.**
>
> We deeply appreciate your positive comments, as well as the thorough and constructive suggestions. Towards your suggestions on further experimental analysis and the writing quality of experimental section, we have performed plenty of revisions and improvements on the experimental section.
>
> For the large dataset of ImageNet, due to the time limit and the resource limit, we are sorry that we could not provide experiments on it, instead, we add three more subsections and provide more quantitative analysis on the current four datasets. We also extend the ATDA method to the PGD-Adversarial Training (PAT) to show how our method can be transferred to iterative attacks.
>
> 1. In subsection 4.3, we performed additional experiments to measure defense efficiency of ATDA in terms of the local loss sensitivity (defined in [1]) to perturbations and the shift of adversarial data distribution with respect to the clean data distribution. Especially on the quantitative analysis on the embeddings, besides plotting the t-SNE embeddings, we report the detailed MMD distances across domains. The results are shown in Table 3. We observe that: SAT and EAT actually increase the MMD distance across domains of the clean data and the adversarial data as compared with NT. In contrast, PRT and ATDA can learn domain invariance between the clean domain and the adversarial domain. Furthermore, our learned logits representation achieves the best performance on distribution discrepancy.
> [1] https://arxiv.org/abs/1706.05394 (ICML 2017).
>
> 2. Following your valuable suggestion, in subsection 4.4, we provide a more thorough analysis to tease out the benefit of each of the various terms added to the loss functions. The results are shown in Figure 3. Thank you.
>
> 3. Although our work is mainly focused on the adversarial training yet efficient FGSM adversary, we show the scalability and flexibility of ATDA on subsection 4.5. We consider to extend the ATDA method to PGD-Adversarial Training (PAT) : adversarial training on the noisy PGD (iterative attack). Meanwhile, we implement an extension of ATDA (called PATDA) by combining adversarial training on the noisy PGD with domain adaptation. As the evaluation shown in Table 4, PATDA exhibits stronger robustness against various bounded adversaries as compared to PAT.
>
>
> We thank you again for the valuable feedback and comments, which have improved the manuscript apparently. We have tried to address most of your concerns and changed the abstract, introduction and conclusion accordingly in the revision. We hope our effort can convince you the quality of our work.

---

### Public Comment · (anonymous) · 2018-11-14
**Comparison against Madry et al. (2018)?**

In Tables 3-6 I would have expected the authors provide a comparison to Madry et al. (2018) which provides the strongest white-box robustness to date. In particular, this paper reports CIFAR-10 numbers at eps=4/255 while Madry et al. uses the harder 8/255. How much robustness is lost by using only one step of FGSM adversarial training?

---

> ### Public Comment · (anonymous) · 2018-11-15
> **Concur**
>
> I concur. This is a necessary baseline.

---

> ### Author Response · Authors · 2018-11-22
> **Addressed by extending our ATDA method to the method of Madry et al.**
>
>
> Thanks for your interest in our paper.
>
> Since (the noisy) PGD can samples more sufficient adversarial examples in adversarial domain, adversarial training on it yields more robust models than adversarial training on FGSM. However, PGD-Adversarial Training (PAT) [1] is challenging to scale to deep or wide neural networks, as it increases the training time by a factor that is roughly equal to the number of PGD steps.
>
> [1] Aleksander Madry, Aleksandar Makelov, Ludwig Schmidt, Dimitris Tsipras, and Adrian Vladu. Towards deep learning models resistant to adversarial attacks. In International Conference on Learning Representations(ICLR), 2018.
>
> Our work is mainly focused on the adversarial training yet efficient FGSM adversary due to its scalability. The empirical evidence shows that our ATDA method has great generalization ability to various adversaries as compared to SAT (adversarial training on FGSM).
>
> To address your concerns, in subsection 4.5, we extend the ATDA method to PGD-Adversarial Training by combining adversarial training on the noisy PGD with domain adaptation. Thus, we implement an extension of ATDA for PAT, called PATDA. The evaluation results suggest that PATDA has stronger robustness on various attacks as compared to PAT. The results indicate that domain adaptation can be applied flexibly to adversarial training on other adversaries to improve the robustness of the model. This experiment exhibits a good transfer ability of our domain adaptation method. Thank you for the nice suggestion.

---

### Author Response · Authors · 2018-11-22
**Paper revision 1**

Dear Reviewers,

We deeply appreciate all reviewers for the thorough comments and valuable suggestions, which definitely help the improvement of our paper! We would like to briefly summarize our modification here and leave specific concerns in individual comments to each of you.

Our main modifications are as follows:

1) For the Experimental Section (section 4), we have reorganized the content to reduce the duplication of expression and further investigate the defense performance of ATDA.
A new subsection 4.3 is added to show the defense efficiency of ATDA in terms of the local loss sensitivity (defined in [1]) to perturbations and the shift of adversarial data distribution with respect to the clean data distribution, as suggested by Reviewer 1 and 2;
    [1] https://arxiv.org/abs/1706.05394 (ICML 2017).

2) A new subsection 4.4 is added to show the ablation studies on ATDA so as to tease out the benefit of each of the various terms added to the loss functions, as suggested by Reviewer 2;

3) To address an anonymous comment, a new subsection 4.5 is added to show the scalability and flexibility of ATDA and the performance of the extension on adversarial training on the noisy PGD (called PATDA) as compared with the original adversarial training on the noisy PGD (called PAT);

We hope all our effort can make our paper more comprehensive and address your concerns. Thank you very much!

Bests,
Authors.

---

### Meta-Review · Area_Chair1 · 2018-12-11
**Using domain adaptation for robust adversarial learning**

**Confidence:** 4
**Recommendation:** Accept (Poster)

**Metareview:**

The paper presents an interesting idea for increasing the robustness of adversarial defenses by combining with existing domain adaptation approaches. All reviewers agree that the paper is well written and clearly articulates the approach and contribution.

The main areas of weakness is that the experiments focus on small datasets, namely CiFAR and MNIST.  That being said, the algorithm is reasonably ablated on the data explored and the authors provided valuable new experimental evidence during the rebuttal phase and in response to the public comment.